# Carbapenem-Resistant Gram-Negative Fermenting and Non-Fermenting Rods Isolated from Hospital Patients in Poland—What Are They Susceptible to?

**DOI:** 10.3390/biomedicines10123049

**Published:** 2022-11-25

**Authors:** Małgorzata Brauncajs, Filip Bielec, Anna Macieja, Dorota Pastuszak-Lewandoska

**Affiliations:** 1Department of Microbiology and Laboratory Medical Immunology, Medical University of Lodz, 90-151 Lodz, Poland; 2Medical Microbiology Laboratory, Central Teaching Hospital of Medical University of Lodz, 92-213 Lodz, Poland

**Keywords:** gram-negative rods, carbapenem resistance, antimicrobial susceptibility

## Abstract

Gram-negative fermenting and non-fermenting bacteria are important etiological factors of nosocomial and community infections, especially those that produce carbapenemases. *Klebsiella pneumoniae*, *Acinetobacter baumannii,* and *Pseudomonas aeruginosa* are the most frequently-detected carbapenemase-producing microorganisms. The predominant type of resistance is metallo-β-lactamase (MBL). These bacteria are predominantly isolated from bronchial alveolar lavage, urine, and blood. Carbapenemase-producing Enterobacterales (CPE) strains are always multi-drug-resistant. This significantly limits the treatment options for this type of infection, extends the time of patient hospitalization, and increases the risk of a more severe and complicated disease course. Preventing the transmission of these microorganisms should be a major public health initiative. New antibiotics and treatment regimens offer hope against these infections.

## 1. Introduction

Gram-negative fermenting and non-fermenting bacteria are extremely important etiological factors of nosocomial and community infections. Recent years have seen an increase in the global prevalence of bacterial strains producing enzymes that hydrolyze carbapenems (carbapenemases): antibiotics previously considered as last resort drugs in the treatment of severe infections caused by Gram-negative microorganisms [1].

Carbapenem resistance may be produced by various mechanisms such as the production of carbapenemases, porin loss, or mutations in efflux pumps, mainly. Carbapenemases are of the greatest clinical importance as they can move between bacteria of the same or different species by horizontal transfer via plasmids, integrons, or transposons [2].

Currently, two criteria are used to classify β-lactamases: the so-called functional system and the structural system [3,4]. Figure 1 shows the location of carbapenemases in both of these classifications.

The first, by Bush and Jacoby [3], is based on a comparison of the rate of hydrolysis of various β-lactams and the susceptibility of β-lactamases to inhibition by some β-lactams (aztreonam, cloxacillin, and β-lactamase inhibitors—clavulanic acid, sulbactam, and tazobactam) and EDTA and NaCl. It is a classification based on functional similarity. It distinguishes four main functional groups, marked with numbers 1 to 4. Group 1 are β-lactamases preferring cephalosporins; they are also active against penicillins and monobactams but inhibited by cloxacillin. Traditionally, they are referred to as “AmpC cephalosporinases.” Group 2 is the most numerous and diverse, divided into 12 subgroups. There are both penicillinases and cephalosporinases, with a narrow, wide, broad, or extremely wide substrate spectrum, covering virtually all β-lactams. The common feature of these enzymes is their susceptibility to inhibition by β-lactam inhibitors (clavulanic acid, tazobactam, and sulbactam), which may also be reduced by mutation. Group 3 consists of metallo-β-lactamases (MBLs), hydrolyzing penicillins, cephalosporins, and carbapenems. MBLs are inhibited by EDTA but not inhibited by β-lactam inhibitors, while group 4 is just a few enzymes hydrolyzing only penicillins and is poorly studied; in the latest version of the functional classification system, this group has been omitted [5,6].

The second system was proposed by Ambler [4]. It is based on the analysis of the amino acid sequence of enzymes and groups of β-lactamases according to their evolutionary relatedness, i.e., the classification is based on their molecular structure. The comparative analysis of the amino acid sequences of β-lactamases allowed one to distinguish four classes of enzymes, designated from A to D. The classes A, C, and D are serine β-lactamases, while MBLs are in the B class. Both divisions of β-lactamases correlate well with each other. All of the enzymes that make up functional group 1 constitute structural class C. Group 2 contains β-lactamases of classes A and D, while group 3 corresponds to class B. The enzymes belonging to functional group 4 have not been structurally characterized. Recently, Guiana extended-spectrum β-lactamase (GES) class A carbapenemases, which are not routinely detected in phenotypic tests and are an essential aspect of the etiology of hospital patients’ infections. Their identification is guaranteed only by tests based on molecular biology methods [7].

Antibiotic resistance continues to grow and is one of the major public health problems globally—mainly due to the excessive and inappropriate use of antimicrobials in humans, animals, and plants. Additionally, in some countries, the lack of infection control has contributed to the spread of drug-resistant pathogens. The recent years associated with the COVID-19 pandemic are also associated with the overuse of antibiotics when treating symptoms of infections coexisting with SARS-CoV-2 infection; it has also accelerated the emergence and spread of antimicrobial resistance [8].

The spread of carbapenemase-producing Enterobacterales (CPE) in Europe (including Poland) has been observed for over a decade. In Poland, the most commonly spread carbapenemases are KPC—*Klebsiella pneumoniae* Carbapenemase (Ambler class A), NDM—New Delhi Metallo-β-lactamase (class B), VIM—Verona Integron-encoded Metallo-β-lactamase (class B), and carbapenemases type-OXA-48 (class D) [9]. Recently, in one of the hospitals in Germany, a new type of MBL was detected in four bacterial isolates taken from patients’ infections, named German metallo-β-lactamase type 1 (GMB-1) [10].

A significant increase in the occurrence of carbapenemase-producing organisms isolated from clinical materials was noted in hospitals in Lodz, Poland from 2014 to 2018. The dominant resistance mechanism was MBL [9].

The aim of the study was to determine the antibiotic susceptibility profiles of carbapenemase-producing Gram-negative rods isolated from clinical materials from patients hospitalized in Lodz, Poland, from November 2021 to May 2022. The obtained results were used to answer the question of whether there is any alternative treatment for critical carbapenem-resistant infections apart from last resort drugs.

## 2. Materials and Methods

A total of 107 strains producing KPC, MBL, and OXA-48 carbapenemases were investigated. All strains were isolated from clinical samples: bronchial alveolar lavage (BAL), blood, urine, rectal swab (CPE screening), lower respiratory specimen (other than BAL), intraoperative swab, nasal swab, wound swab, and pressure ulcer swab.

All bacteria were stored in Viabank^TM^ storage beads (Medical Wire and Equipment, Corsham, UK) at −80 °C maximum for six months and regenerated on Columbia Agar with 5% sheep blood (Thermo Fisher Scientific, Waltham, MA, USA) for 18–24 h at 37 °C. The strains were subjected to biochemical identification and drug susceptibility assessment using an automated BD Phoenix system (Becton Dickinson and Company, Franklin Lakes, NJ, USA). Colistin sensitivity was assessed using the MICRONAUT MIC-Strip colistin assay (MERLIN Diagnostika, Bornheim-Hersel, Germany).

The ability of all studied strains to produce carbapenemases was assessed using a biochemical diagnostic test (CIM, carbapenem inactivation method) [11] and then using the phenotypic methods, according to EUCAST 2022 [12] and the Polish National Reference Centre for Microbial Susceptibility (KORLD) [13].

The presence of common carbapenem resistance mechanisms (KPC, OXA-48, NDM, and VIM) in invasive isolates has been confirmed by PCR in the Polish National Reference Centre for Microbial Susceptibility (KORLD), while the presence of GES gene has been studied in Department of Microbiology and Laboratory Medical Immunology, Medical University of Lodz, with a positive control strain for tested mechanism confirmed previously by the Polish National Reference Centre for Microbial Susceptibility (KORLD). Bacterial genomic DNA was obtained using the Genomic Mini AX Bacteria Spin kit (A&A Biotechnology, Gdansk, Poland), according to the manufacturer instruction. DNA was amplified using HS PCR Kit 1 (A&A Biotechnology, Gdansk, Poland). PCR products were analyzed by electrophoresis on a 2% agarose gel. As a DNA marker, we used GeneRuler 100 bp DNA Ladder (Thermo Fisher Scientific, Waltham, MA, USA).

Descriptive statistics were prepared using Microsoft Excel 2019 software (Microsoft Corporation, Redmond, WA, USA).

### Ethical Issues

The presented research was conducted with the high ethical standards. The study involved only anonymized records, without the possibility of identifying a specific human being. All bacterial strains were previously secured in the culture collection of our research unit, using consecutive code identification numbers. The only clinical data concern the sex and age of the patients and the type of biological material from which the bacterial strain was isolated.

## 3. Results

A total of 107 strains of Gram-negative rods were analyzed. These were isolated from 102 patients (49 women and 53 men) aged 1 to 92 years (mean age 62 years). Among these, 70 strains were found to produce one carbapenemase, 36 were found to produce two carbapenemases (14 strains had both NDM and GES, 14 strains had both unspecified MBL and GES, 5 strains had both KPC and OXA-48, 1 strain had both KPC and GES, and 1 strain had both unspecified MBL and OXA-48). Figure 2 and Figure 3 present the distribution of clinical samples and bacterial species analyzed in conducted research.

Among the tested bacterial strains producing carbapenemases, 58 MBL-positive strains (27 NDM, 2 VIM, and 29 with no molecular identification), 6 OXA-48-positive strains, 13 KPC-positive, and 38 GES-positive strains were identified. In the case of the remaining 25 isolates, neither phenotypic nor molecular tests confirmed the presence of KPC, OXA-48, MBL, GES, NDM, or VIM mechanisms. Figure 4 presents the distribution of carbapenemase-producing mechanisms among the analyzed strains.

Among carbapenemase-producing microorganisms, the most common Enterobacterales were found to be *K. pneumoniae* and *Escherichia coli*, together with two species of non-fermenting Gram-negative rods, i.e., *Acinetobacter baumannii* and *Pseudomonas aeruginosa*. In *K. pneumoniae*, *E. coli*, and *P. aeruginosa*, the dominant type of identified resistancemechanism was GES, but collectively, MBLs made up the largest group. A detailed susceptibility analysis was performed for these four species, as shown in Figure 5, Figure 6, Figure 7 and Figure 8, respectively.

The susceptibility testing of *K. pneumoniae* isolates showed that 100% of the tested strains were resistant to amoxicillin with clavulanic acid, piperacillin, piperacillin with tazobactam, cefepime, ceftazidime, cefotaxime, cefuroxime, and levofloxacin. In total, 98% of strains showed resistance to imipenem, 96% to meropenem, and 91% to ertapenem. In addition, 97% of *K. pneumoniae* isolates were resistant to aztreonam and 98% to ciprofloxacin. The resistance to aminoglycosides turned out to be less common than to carbapenems: 78% of strains were resistant to amikacin and 76% to gentamicin; only in the case of tobramycin did the value exceed 90%. It was also found that 27% of the isolates were resistant to fosfomycin, an antibiotic belonging to phosphonic acid derivatives, and 62% were resistant to trimethoprim/sulfamethoxazole. Colistin, often the only alternative in the treatment of infections caused by multidrug-resistant Gram-negative bacilli, turned out to be ineffective in vitro for 16% of *K. pneumoniae* isolates.

Among the *E. coli* isolates, 100% of the tested strains were found to be resistant to ampicillin, amoxicillin with clavulanic acid, piperacillin, piperacillin/tazobactam, cefepime, ceftazidime, cefuroxime, ertapenem, aztreonam, ciprofloxacin, levofloxacin, tigecycline, and trimethoprim/sulfamethoxazole. In addition, 71% of the strains were found to be resistant to imipenem and meropenem, 29% to aminoglycosides, and 14% to fosfomycin. However, all tested microorganisms were sensitive to nitrofurantoin and colistin.

Among the non-fermenting Gram-negative rods, *A. baumannii* isolates showed 100% resistance to carbapenems; in addition, 96% of the strains were resistant to trimethoprim/sulfamethoxazole and fluoroquinolones; more than 80% were insensitive to amikacin and tobramycin; 46% to gentamicin; and 8% to the drug of last resort, colistin.

In the case of *P. aeruginosa*, 100% resistance to piperacillin and imipenem was observed. Additionally, 64% of the strains were resistant to meropenem, piperacillin/tazobactam, and amikacin; 79% to the cephalosporins cefepime and ceftazidime; 71% to ciprofloxacin; 82% to levofloxacin; 20% to aztreonam; and 57% to tobramycin. In addition, 14% of the isolates were resistant to the drug of last resort, colistin.

## 4. Discussion

The strains considered in our study came from several hospital laboratories in Łódź, central Poland. Unfortunately, we did not have access to epidemiological data from other laboratories, but based only on the data from our hospital, in 2021, among all of the biological material routinely tested, carbapenem-resistance was found in 6% of Enterobacterales, 8% of *Pseudomonas* spp., and 16% of *Acinetobacter* spp. Considering CPE screening tests only, 40% of Enterobacterales, 25% of *Pseudomonas* spp., and 15% of *Acinetobacter* spp. showed resistance to carbapenems. Additionally, considering only invasive isolates (from blood cultures), carbapenem-resistance was found in 2% of Enterboacterales, 20% of *Pseudomonas* spp., and 9% of *Acinetobacter* spp. These results are significantly lower than those presented for Poland in the WHO report (data from 2016–2020) [14], especially when comparing the *Acinetobacter* genus—10 times less frequent carbapenem-resistance in our hospital than in all of Poland.

Our research reveals a higher percentage of *K. pneumoniae* MBL strains compared to other collected isolates. Similarly, Albiger et al. [15] reported a higher frequency of MBL enzymes compared to KPC in a study of carbapenemase-producing *Enterobacteriaceae* in Europe. Additionally, van Duin et al. [16] have noted that Poland appears to have one of the highest prevalances of MBL enzymes in Europe.

The strains producing carbapenemases have been reported to be among the most common causes of respiratory and urinary tract infections, and of systemic infections such as bacteremia and sepsis [17,18,19]. As such, these bacteria are commonly isolated from bronchial alveolar lavage (BAL), urine, and blood [17]. Similar observations were reported in analyses of nosocomial infections caused by carbapenemases-producing *Enterobacteriaceae* [18,19].

In one study, isolates resistant to carbapenems were identified, mainly from urine (25.9%) and from secretions from the lower respiratory tract (14.3%) and blood (17%) [19]. In the present study, most CPE strains came from diagnostic tests, indicating a high degree of colonization of the gastrointestinal tract. Such colonization is unfavorable because of the possibility of the transmission of these bacteria to other patients or medical personnel, and their further spread in the hospital environment [20].

The Gram-negative rod most commonly identified worldwide as CPE is *K. pneumoniae* [16,21,22], which was also confirmed in our study. Carbapenemase-producing *K. pneumoniae* strains present a significant clinical problem due to their increased capacity for patient colonization, epidemic potential, and high antibiotic resistance [23]. Our present findings indicated that such strains were most susceptible to colistin (84%). Iovleva et al. [24] also showed similar resistance to amikacin (70%) and lower resistance to meropenem (80%), imipenem (83.3%), tobramycin (90%), and ciprofloxacin (93.3%).

Many strains producing carbapenemases demonstrate multidrug resistance (MDR), i.e., simultaneous insensitivity to at least one antibiotic from three or more drug classes used to treat infections caused by a given group of bacteria [25]. It has been proposed that MDR significantly reduces the treatment options and can lead to therapeutic failure [26]. One option for countering MDR carbapenemase-producing bacilli is the use of combination therapy. Fritzenwanker et al. [27] propose that aminoglycoside with meropenem may be a possible therapeutic option for strains sensitive to aminoglycosides, while fluoroquinolones with meropenem may be suitable for strains showing sensitivity to ciprofloxacin. It has also been proposed that colistin may be suitable for strains resistant to aminoglycosides, and tigecycline with meropenem for those sensitive to either colistin or tigecycline [28].

Most carbapenemase-producing *E. coli* and *K. pneumoniae* strains are still sensitive to polymyxins (polymyxin B and colistin), as confirmed by our present data (100% *E. coli*-susceptible isolates; 84% *K. pneumoniae*-susceptible isolates). As such, polymyxins may represent a suitable last resort for the systemic therapy of serious infections; however, occasional toxicity, especially nephrotoxicity, limits their use [29]. Among the non-fermenting Gram-negative rods, the most common producers of carbapenemases are *A. baumannii* (62.6%) and *P. aeruginosa* (26.1%); these confirm previous findings [30]. In the present study, 8% of *A. baumannii* isolates and 14% of *P. aeruginosa* isolates were resistant to colistin, the drug used in Poland for treating CPE infections. Similarly, Viehman et al. [31] reported that 5.3% of *A. baumannii* strains also showed resistance to colistin.

For multi-drug resistant Gram-negative bacilli, polymyxins B and E are recognized as a life-saving therapeutic option in the absence of other treatments. Unfortunately, in 2015, the mobilized colistin resistance type 1 gene (MCR-1) carried by plasmids was discovered in the bacterium *E. coli* in China [32]. Subsequent research around the world led to the discovery of different variants of the MCR gene [33]. An alternative in this case is a combination therapy, e.g., colistin with tigecycline and new antibiotics such as ceftazidime with avibactam starting to play an important role in the treatment of *K. pneumoniae* MDR infections, including colistin-resistant isolates that produce KPC [34,35].

Most carbapenem-resistant Enterobacterales strains in our study were susceptible to fosfomycin—more than 80% of *E. coli* and more than 70% of *K. pneumoniae*. Fosfomycin has regained interest for its role in the treatment of severe infections sustained by resistant strains (e.g., *K. pneumoniae* KPC) in combination with other drugs [36]. Currently, it is registered only for urinary tract infections in Poland, but its off-label use is wide-ranging, such as neurological infections, orthopedic infections, or sepsis [36,37]. Fosfomycin is chemically different from all other antibacterial drugs—it is actively transported inside bacterial cells. Thanks to the smallest particle size among known antibiotics, it easily overcomes barriers in the body. It has a broad spectrum of bactericidal activity against numerous Gram-positive and Gram-negative bacteria [38,39]. The clinical data analyzed by Putensen et al. [36] pointed out that fosfomycin could be an effective solution for severe bacterial infections in critically ill patients due to its pharmacokinetic properties and the high percentage of strains susceptible to it. In recent papers, Singkham-In et al. [40,41] observed in vitro activity of fosfomycin with azithromycin against carbapenem-resistant *K. pneumoniae* and *A. baumannii* clinical isolates. Perhaps that would be the right therapy to fight MDR Gram-negative rods. This topic requires further research and clinical observations.

However, new combinations of antibiotics in conjunction with β-lactamase inhibitors e.g., meropenem with vaborbactam or imipenem with relebactam, offer hope in treating resistant strains [42,43,44,45,46,47]. Additionally, some new agents are developed to cope with such infections [48,49,50]. Each new and effective antibiotic is worth its weight in gold due to the progressive development of bacteria resistant to the antimicrobial drugs used so far. The Review on Antimicrobial Resistance [51] predicts that in 2050, the number of deaths from drug-resistant infections will increase from 700,000 to 10 million per year. Multi-drug resistant strains are one of the greatest threats to humanity.

In a special report from 2017 [52], WHO presented a list of bacteria for which pursuing a discovery of new antimicrobials was a priority. One of the most important aspects of this research was the development of new antimicrobials effective against CPE. Since the publication of this report, the European Medicines Agency (EMA) has approved several new broad-spectrum antibiotics. Two of them are new combinations of carbapenems with non-β-lactam inhibitors of β-lactamases (also active against carbapenemases), belonging to two new groups of inhibitors: diazabicyclooctane (relebactam, associated with imipenem) and boron (vaborbactam, associated with meropenem) [47]. The third new drug is a siderophore cephalosporin (cefiderocol) with an innovative mechanism of penetration into the bacterial cell [48]. Another antibiotic is a new tetracycline (eravacycline) [49]. The last innovative antibiotic is lefamulin—the first pleuromutilin approved for general use for humans [50]. New registrations increased the number of available therapeutic options in the treatment of complicated and other infections:-the urinary tract, including pyelonephritis (meropenem/vaborbactam, cefiderocol),-complicated intra-abdominal infections (meropenem/vaborbactam, eravacycline),-nosocomial pneumonia, including ventilator pneumonia (meropenem/vaborbactam, imipenem/relebactam, and lefamulin).

These drugs can also be used for patients who have a blood infection related to any of the above-mentioned infections; this is very important when such a relationship is suspected.

### Study Limitations

In total, 107 strains were tested in the study, which may be considered a small number. However, the strains came from several hospital centers in a large city in the center of Poland, and the carbapenem-resistant isolates are relatively rare compared to all specimens processed in local microbiology medical laboratories. The total number of strains resulted in small numbers of isolates of specific species; therefore, the antimicrobial susceptibility was analyzed only for the four most frequently identified species.

Another limitation was the method of fosfomycin resistance testing. According to the current EUCAST recommendations [53], the reference method for the determination of fosfomycin susceptibility is agar dilution. For the purposes of this publication, determinations with the BD Phoenix automated microbiological system were sufficient in the authors’ opinion. However, it is important to note that the targeted antibiotic therapy with fosfomycin can only be based on an antibiogram prepared in accordance with the current standards. Susceptibility to all other tested antimicrobials was done following the standards.

## 5. Conclusions

The production of enzymes hydrolyzing β-lactam antibiotics is undoubtedly one of the most dangerous mechanisms of conferring resistance to antibacterial drugs. The most crucial problem is the spread of Gram-negative rods that produce carbapenemases. Our findings indicate that all tested carbapenemase-producing strains showed multidrug resistance. This feature significantly limits the treatment options for this type of infection and prevents effective control over their abundance and spread, particularly in the hospital environment. The resulting infection can extend hospitalization time, increase the risk of more severe and complicated disease courses, and increase healthcare cost and mortality. Hence, it is essential to monitor the results of bacteriological tests globally in the hospital facilities to detect multidrug-resistant bacteria early, prevent their spread, and develop appropriate strategies for all forms of antimicrobial therapy in hospitals.

## Figures and Tables

**Figure 1 biomedicines-10-03049-f001:**
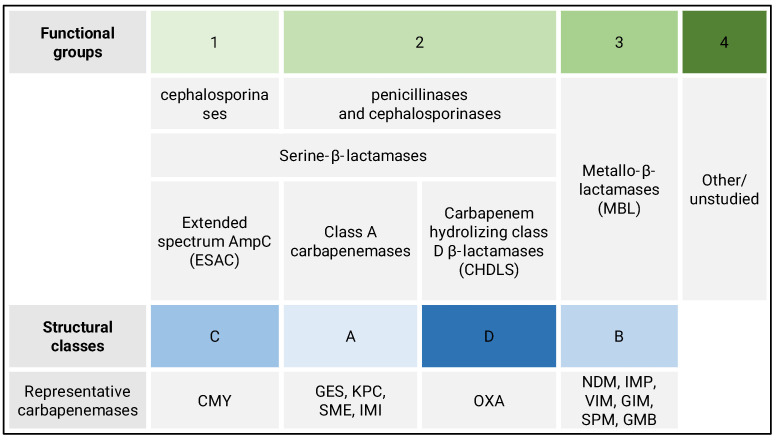
Comparison of β-lactamases classification systems with representative carbapenemases (AmpC—ampicillin chromosomal cephalosporinase, CMY—cephamycin-hydrolyzing β-lactamase, GES—Guiana extended-spectrum, KPC—*Klebsiella pneumoniae* carbapenemase, SME—*Serratia marcescens* enzyme, IMI—imipenem-hydrolyzing β-lactamase, OXA—oxacillinases, NDM—New Delhi metallo-β-lactamase, IMP—imipenemase β-lactamase, VIM—Verona integron-encoded metallo-β-lactamase, GIM—German imipenemase, SPM—São Paulo metallo-β-lactamase, and GMB—German metallo-β-lactamase).

**Figure 2 biomedicines-10-03049-f002:**
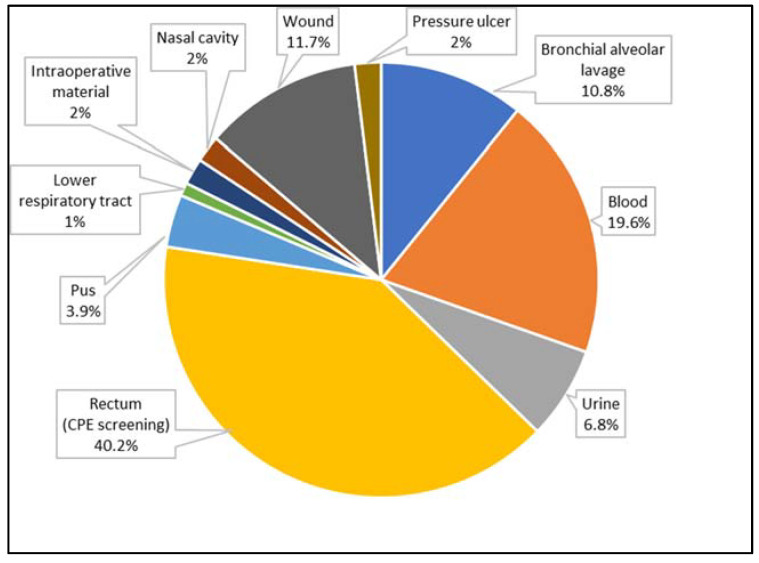
The percentage of biological materials from patients in hospitals in Lodz (Poland) taken under consideration in this study.

**Figure 3 biomedicines-10-03049-f003:**
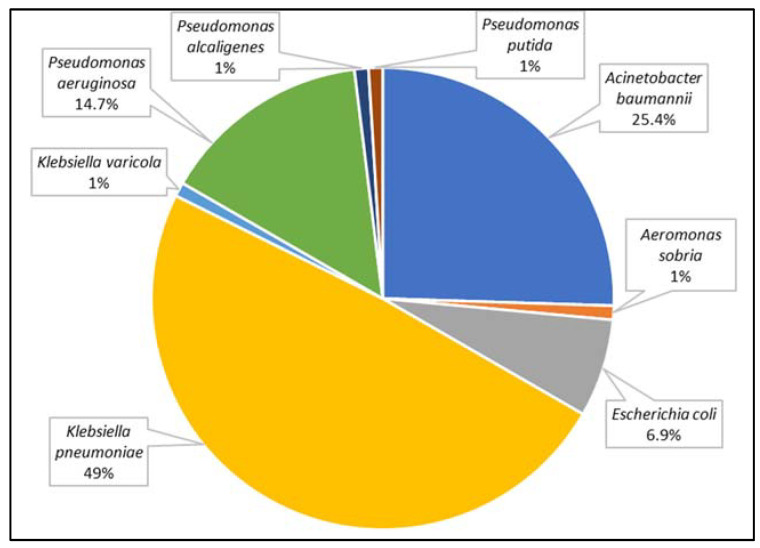
The percentage of carbapenemase-producing species identified from patients hospitalized in Lodz (Poland) taken under consideration in this study.

**Figure 4 biomedicines-10-03049-f004:**
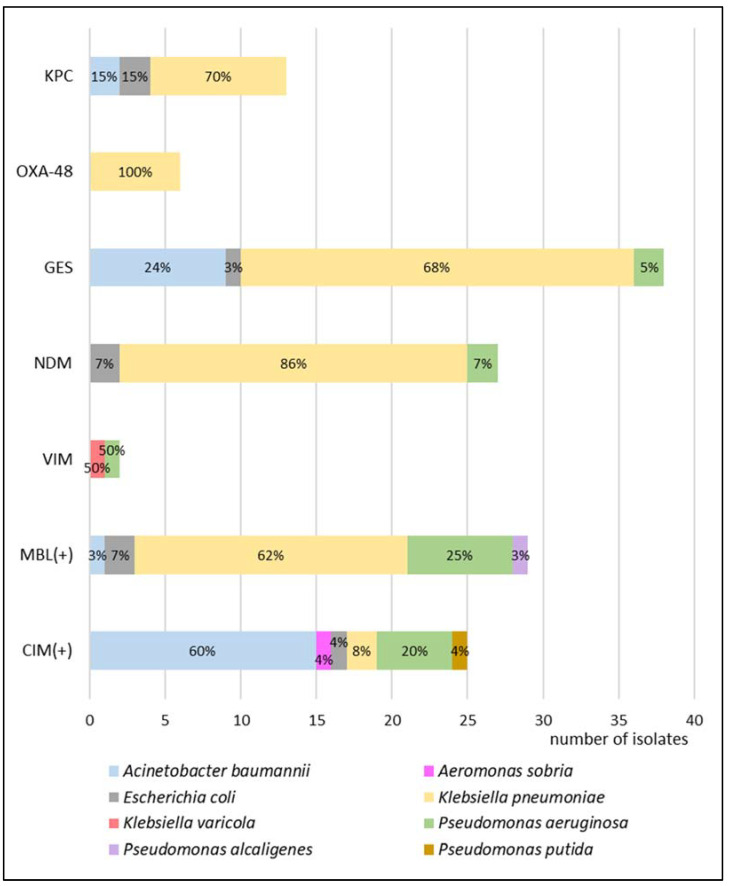
The percentage of bacteria of each species that produce certain types of carbapenemases taken under consideration in this study. The presence of KPC, OXA-48, GES, NDM, and VIM were confirmed by the molecular method. MBL(+) means the positive result of phenotypic test but no molecular identification. CIM(+) means the positive carbapenem inactivation test but no phenotypic or molecular identification (KPC—*Klebsiella pneumoniae* carbapenemase, OXA-48—oxacillinase-48, GES—Guiana extended-spectrum, NDM—New Delhi metallo-β-lactamase, VIM—Verona integron-encoded metallo-β-lactamase, MBL—metallo-β-lactamase, and CIM—carbapenem inactivation method).

**Figure 5 biomedicines-10-03049-f005:**
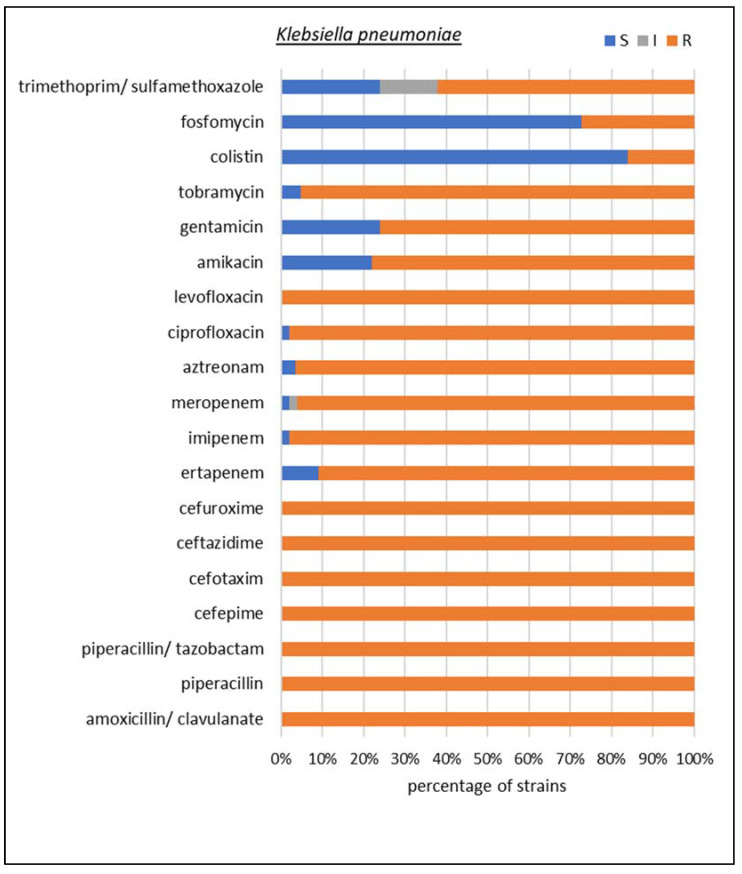
Drug susceptibility of the studied *Klebsiella pneumoniae* strains producing carbapenemases (*n* = 50).

**Figure 6 biomedicines-10-03049-f006:**
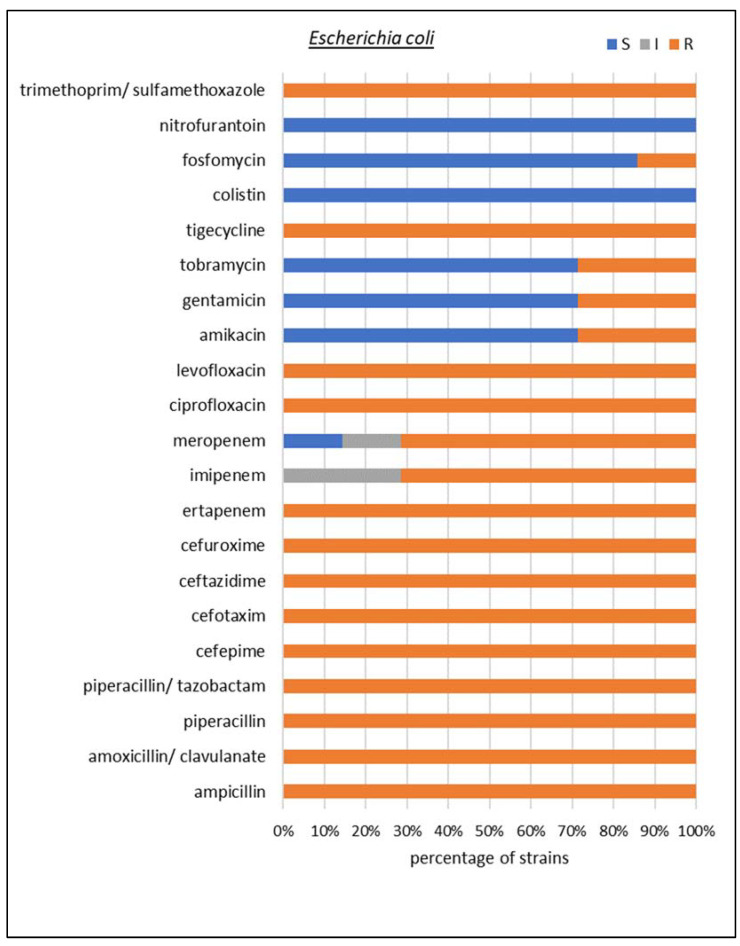
Drug susceptibility of the studied *Escherichia coli* strains producing carbapenemases (*n* = 7).

**Figure 7 biomedicines-10-03049-f007:**
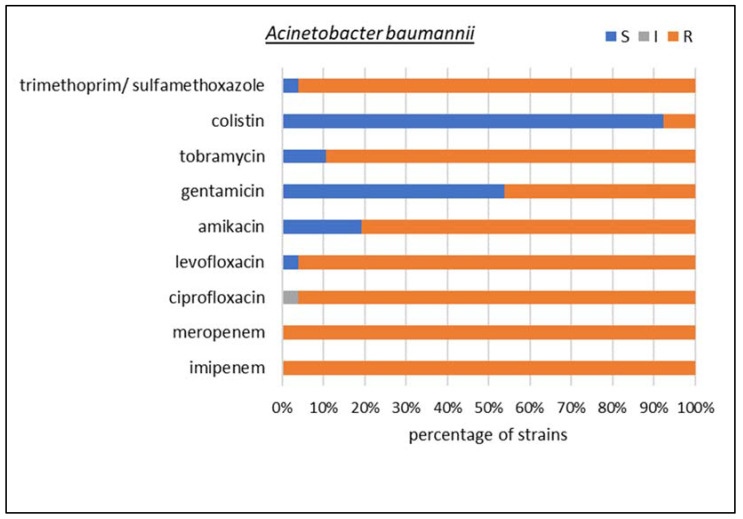
Drug susceptibility of the studied *Acinetobacter baumannii* strains producing carbapenemases (*n* = 26).

**Figure 8 biomedicines-10-03049-f008:**
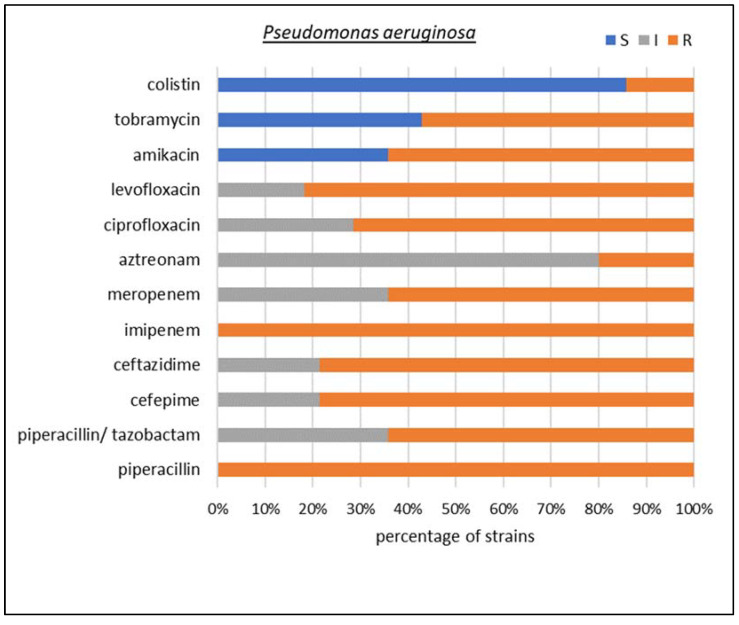
Drug susceptibility of the studied *Pseudomonas aeruginosa* strains producing carbapenemases (*n* = 15).

## Data Availability

Not applicable.

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
