# Peer review of "Carbapenem-Resistant Gram-Negative Fermenting and Non-Fermenting Rods Isolated from Hospital Patients in Poland—What Are They Susceptible to?"

_biomedicines, 2022, doi:10.3390/biomedicines10123049_

Round 1

Reviewer 1 Report (Previous Reviewer 1)

Thank you for considering my recommendations. I think that the present manuscript can be accepted.

Author Response

We are very grateful for the suggestions made earlier. Applying them to the manuscript made the work more valuable.

Reviewer 2 Report (New Reviewer)

The manuscript from Bruncajs and colleagues described the epidemiology of Carbapenemase Producing Gram Negative Rods (CPGNR) in a single centre in Poland.

Albeit interesting, this manuscript cannot be published without corrections and considerations of the comments below.

Major comments:

Methods : The reader cannot understand how strains were selected in this study. The authors stated the strains were isolated from infections (line 152), however 40% were taken from the rectum of patients (screening). The methods part should be rewritten and the authors should consider all Gram Negative Rods cultivated during the study period and tell how many were selected in case of presumed Carbapenemase production according to phenotypic Eucast methods. This would give to the reader the percentage of CPGNR among all strains of this center. Then the authors should say how many were selected from sterile tissues or liquids (infection strains) and how many were tested using carbapenem inactivation methods (CIM) before molecular detection of of KPC, OXA-48, NDM, VIM and GES.

Results : It is impossible for the reader to understand the exact repartition of strains described by the authors. The sum of strains is not exact (line 157).  The figures are confusing and the authors should prefer large tables with all strains, and give precise number in the text. How many strains tested positive by CIM have a molecular detection of KPC, OXA-48, NDM, VIM and GES? Then the authors should show a table with phenotypic sensitivity percentages and another table with the percentage of each enzyme for all species each time in a single table. This would be easier to read than figures which are sometimes duplicated.

Discussion and conclusion : far too long. Discussion and conclusion should focus on the results of the study.

Minor comments:

Results: please explain line 178

Round 2

Reviewer 2 Report (New Reviewer)

It is unfortunate that the authors could not even estimate to which percentage of all strains isolated in their hospital network do these study strains correspond to? It would have given an estimation of how frequent were these kinds of strains in routine.

Minor comments :

Pseudeomonas figure 8

line 301 : a "new" should be removed

For information, the duplicate figure I was talking about.

Author Response

We kindly thank you for your comments and suggestions. We were able to quickly obtain epidemiological data on carbapenem-resistant isolates in our hospital for the previous year. We've included these data in the manuscript. Minor fixes have also been made.

This manuscript is a resubmission of an earlier submission. The following is a list of the peer review reports and author responses from that submission.

Round 1

Reviewer 1 Report

Major comments

Materials and methods

1.    Line110-112: I am surprised that the authors describe the utilization of 107 strains producing KPC, MBL and OXA-48 carbapenemases but these only were detected by phenotypic methods and non-molecular methods (i.e. PCR) were used. It would be convenient to confirm which type of carbapenemase was present in these clinical isolates.

2.    How Fosfomycin was tested? EUCAST recommended agar dilution to perform susceptibility testing of Fosfomycin. If not is the case, add this in limitations.

Results

1.    Line 119. What kind of carbapenemases were detected among the 8 isolates with two different carbapenemases?

2.    Line 123. It would be fantastic if were possible to test uncommon carbapenemases (GES, SPM, SIM, GIM, etc.). If this is not possible, it would be fine to perform, at least, a PCR of the most common carbapenemases (KPC, IMP, VIM, NMD, OXA-48) in the remaining 33 isolates in which phenotypic test were negative but CIM positive.

3.    Line 183. Carbapenemases usually confer resistance to piperacillin/tazobactam. It is surprisingly that 14% of E. coli with carbapenemase production were susceptible to piperacillin/tazobactam. In the same line, in the case of P. aeruginosa, only 64% of the strains were resistant. Please, could the authors explain which kind of carbapenemases were detected in those isolates susceptible to piperacillin/tazobactam?

Discussion

1.    Line 273-298: these lines can be removed. This is a single study, not a revision work.

2.    A “limitations of the study” paragraph is lack. Please, add it taking in consideration the several limitations of the study.

Minor comments

Introduction

1.    Line 29: reformulate the sentence to “Carbapenem resistance may be produced by various mechanisms such as production of carbapenemases, porin loss or mutations in efflux pumps, mainly”.

2.    Line 32: remove “of” in the sentence: “carbapenemases are of the most…”

3.    Line 44: remove “weakly susceptible to inhibition with clavulanic acid,

4.    Figure 1: TEM and SHV enzymes are not carbapenemases. Please, remove it.

5.    Line 83: Enterobacterales must be in italics

Materials and methods

1.    Line 98 and 120: change “materials” to “samples”

2.    Line 99: Lower respiratory SWABS sounds puzzling. Maybe the authors wanted to say “Lower respiratory TRACT”?

Results

1.    Line 135: “Enterobacterales” must be in italics.

2.    Line 135: Replace Klebsiella pneumoniae by K. pneumoniae

3.    Line 135: Add the percentage of carbapenemases in each microorganism

Discussion

1.    Line 242. Currently, there are many types of mcr gene described, that confers resistance to colistin. Please, reformulate the sentence.

Reviewer 2 Report

Throughout – Significant figures to the ones place should suffice here (e.g. “1%” instead of “0.98%”).  The word “swab” can be dropped from each of the tags with that word.

Throughout, “gram” should be “Gram” (i.e. capitalized) as it is a proper noun.

Abstract – Please spell out the acronyms MBL and CPE.

Line 27 – Please elaborate on what is meant by “previously” here.  Two years, five years, two decades?

Line 30 – What is meant by “outer sheaths”?  Is this the plasma membrane?  Peptidoglycan layer?  Something else?

Lines 32-33 – perhaps “be moved” could be changed to “move”.

Figure 1 – I really like this table!  There is no Bush & Jacoby Group 4 in it, though.  I realize you state that few studies exist and this group has been dropped, but at least a place-holder in the table that says “other”/“unstudied” would be nice.  It would also be nice if you indicate that Bush & Jacoby is functional whereas Ambler is structural.  We should depart from eponymous naming schemes as they are nonlogical and unhelpful, presenting a burden to understanding.  I strongly advise the authors consider dropping the names and adding only “functional” and “structural”.

Line 53 – The word “identified” seems not to make sense here.

Line 58 – “aminoacid” is two words (i.e. “amino acid”).

Line 60 – “is” should be “are” as this is referring to MBL, which is a plural noun.

Lines 63-65 – The authors do not clearly explain why GES is so critical to detect.  Also, this acronym should be spelled out with its first use here.

Line 79 – Perhaps the word “organisms” or “pathogens” would be better than “infections”, since it is the organism that spreads, not the infection.

Line 89 – Spell out GMB.

Lines 91, 120 – What are “clinical materials”?  Is this a clinical isolate from human tissues?  Is “clinical materials” supposed to mean “source of bacterial isolate”?  Please change this term.

Line 92 – The authors provide the beta-lactamase, but not the bacterial species that are prevalent.  What were the bacterial species isolated?

Lines 93-95 – The authors state the aim but what was the goal?  WHY were they studying antibiotic susceptibility profiles?

Figure 2 – “Urina” should be “Urine”.

Figure 3 – Where is Figure 3?  Also, I see a reference to Table 3 in the Figure 2 legend but there is no Table 3, let alone Tables 1 and 2.

Figure 4 – it is difficult to tell the difference between Klebsiella varicola and Acinetobacter baumanniibecause the colors are so similar.  Consider coloring one of them white with black borders.  The label “OX-48” should be “OXA-48”.  Perhaps rather than concentric circles, a stacked column plot would be more useful, showing the actual number of isolates on the x-axis with percentages listed for each organism on the column.

Line 123 – 33 isolates is a third of the sample.  Is there no further evaluation of these strains?  This seems important to investigate.

Lines 124-125 – The method used for identification is not necessary to state here as it is already stated in the Methods section.

Line 135 – “found to Klebsiella” should be “found to be Klebsiella”.

Lines 137-138 – This claim is not supported by Figure 4 in its current format.  Consider revising as suggested.

Figures 5-8 – Please state the number of isolates for each organism that were tested, rather than just percentages.  If these figures were converted into horizontal bar graphs, the names of the antibiotics could be fully written without abbreviation.  Please put a title above the plots with the name of the organism being evaluated.

Lines 196-197 – This statement seems inaccurate as 100% of A. baumannii strains were resistant to carbapenems whereas <100% of K. pneumoniae strains were resistant to carbapenems.

Lines 197-199 – It is unclear how this sentence relates to the prior claim.  In other words, what is similar about the two statements?

Lines 199, 205 – Is it “Enterobacteriaceae” or “Enterobacterales”?

Line 200 – What is meant by “densities”?  Is this a frequency of occurrence?

Lines 205-206 – Either remove the dash between these three words or rewrite as “carbapenemase-producing Enterobacterales”

Line 208 – Again, significant figures to the ones place should suffice here (e.g. “26%” instead of “25.9%”).

Line 209 – “screening tests” as opposed to what?  Diagnostic tests?  A specific test?

Lines 210-212 – Do the authors mean to combine two statements here?   i.e. “Such colonization is unfavorable AND MAY BE due to the possibility…” instead of “Such colonization is unfavorable due to the possibility…”

Lines 213-215 – Is it really just K. pneumoniae strains, though?  Wouldn’t the other three strains detected also fit in this category?  Why single out K. pneumoniae?  Aren’t all patients colonized with E. coli?

Line 219 – “carbapenemase beta-lactamase” should be either “carbapenemase” or “beta-lactamase”.

Lines 230-232 – In the data, K. pneumoniae, A. baumannii, and P. aeruginosa all demonstrate some isolates with resistance to colistin.  That should be reflected in this statement.

Lines 232-234 – Are the authors suggesting that colistin be prescribed to all patients with serious infections?  I do not think that would be a good idea.

Lines 241-243 – Please cite.

Lines 243-244 – In order to state that this is level is high, it is necessary to include a comparator that is not.  Please include such a comparator (i.e. mortality of hospitalized patients infected with carbapenem-sensitive and colistin-sensitive strains).

Line 251 – The authors mean to use “e.g.”, not “i.e.”.

Line 252 – “wide” should be “wide-ranging”.

Line 262 – It is unclear what the authors mean by the word “caliber”

Line 263 – The authors introduce extensively drug-resistant without defining it like they did for MDR.  It would be helpful to write out the distinction between the two.

Line 274-278 – The syntax of this sentence needs some attention.

Line 275 – there are other anti-infectives besides antibiotics (e.g. phages, antibodies, defensins, TLR4 modulators, etc.).  Consider changing the word “antibiotics” to “antimicrobials”, “antibacterial drugs”, or “anti-infectives”.

Line 276 – Italicize “Pseudomonas aeruginosa”.

Line 279 – Please spell out MRSA, VRSA, and VRE.

Line 305 – What is meant by the word “numbers”?  Abundance?

Line 308 – Do the authors mean “antibiogram” instead of “bacteriological profiles”?

Round 2

Reviewer 1 Report

Dear authors, 

I appreciate the changes made to the manuscript. However, without molecular detection by PCR that confirms the type of carbapenemase present in those isolates you can only suggest that the isolate has a class A, B or D carbapenemase. I think that this is crucial in a prevalence study of carbapenemase like this. I encourage you to do it for future works.

Reviewer 2 Report

I'm pleased with how seriously the authors took my suggestions and feel that the modifications they made in response are quite adequate.